# Site-Search Process for Synaptic Protein-DNA Complexes

**DOI:** 10.3390/ijms23010212

**Published:** 2021-12-25

**Authors:** Sridhar Vemulapalli, Mohtadin Hashemi, Yuri L. Lyubchenko

**Affiliations:** Department of Pharmaceutical Sciences, College of Pharmacy, University of Nebraska Medical Center, Omaha, NE 68198-6025, USA; sridhar.vemulapalli@unmc.edu (S.V.); mohtadin.hashemi@unmc.edu (M.H.)

**Keywords:** high-speed AFM, SFiI, site-search, threading, site-bound segment transfer, synaptic complexes

## Abstract

The assembly of synaptic protein-DNA complexes by specialized proteins is critical for bringing together two distant sites within a DNA molecule or bridging two DNA molecules. The assembly of such synaptosomes is needed in numerous genetic processes requiring the interactions of two or more sites. The molecular mechanisms by which the protein brings the sites together, enabling the assembly of synaptosomes, remain unknown. Such proteins can utilize sliding, jumping, and segmental transfer pathways proposed for the single-site search process, but none of these pathways explains how the synaptosome assembles. Here we used restriction enzyme SfiI, that requires the assembly of synaptosome for DNA cleavage, as our experimental system and applied time-lapse, high-speed AFM to directly visualize the site search process accomplished by the SfiI enzyme. For the single-site SfiI-DNA complexes, we were able to directly visualize such pathways as sliding, jumping, and segmental site transfer. However, within the synaptic looped complexes, we visualized the threading and site-bound segment transfer as the synaptosome-specific search pathways for SfiI. In addition, we visualized sliding and jumping pathways for the loop dissociated complexes. Based on our data, we propose the site-search model for synaptic protein-DNA systems.

## 1. Introduction

A critical step in fundamental genetic processes is the interaction between distant DNA regions. This step is controlled by specialized proteins or protein complexes which are responsible for forming site-specific protein-DNA synaptic complexes (i.e., synaptosomes) [1]. The formation of a synaptic complex is a general phenomenon found in gene regulation (e.g., the GalR repressor) [2], site-specific recombination (e.g., Flp, Cre recombinases) [3], and various eukaryotic genome rearrangement systems, such as transposons, the Variable Diversity Joining (V(D)J) recombination system involving tetrameric RAG1/2 assembly [4], and HIV integration systems [5,6]. Notably, type II DNA restriction enzymes form synaptic complexes to complete site-specific DNA cleavage [7]. Still, little is known regarding the molecular mechanisms that underlie how such proteins search for DNA sites during the formation of synaptosomes [1,3,7,8]. A seminal theoretical model by Berg, von Hippel, and Winter (i.e., the BHW model) [9,10,11,12,13] describes simple processes with one-site searches, including sliding, jumping, and intersegmental transfers as the primary pathways. However, these specific pathways cannot explain the assembly of synaptosomes for which one protein holds two or more specific sites on the DNA substrate. If the specific sites needed for the formation of the synaptosome are located on the same DNA molecule (i.e., pairing in *cis*), the search for the two sites leads to the formation of DNA loops. The formation of such loops for various synaptosomes has been visualized with Atomic Force Microscopy (AFM) in our previous papers [2,14,15,16,17,18,19]. Note article [18], in which the assembly of the synaptosome with three DNA binding sites held by the EcoRII restriction enzyme was directly visualized. We also applied time-lapse, high-speed AFM (HS-AFM) for this synaptic system to visualize the dynamics of synaptic complexes assembled by EcoRII [20]. In addition to sliding and jumping when bound to a single site, we visualized the growth of looped complexes, in which the protein bound two separate DNA segments and searched by threading the DNA. This mechanism is specific for synaptic protein-DNA complexes, and poses the following questions: What is the contribution of this pathway to the site-search? Does the search mechanism for the synaptic system utilize a site-transfer pathway? How do they all contribute to the entire site-search process?

To address these questions, we selected the SfiI restriction enzyme as our model system. SfiI is a type IIF restriction enzyme and assembles as a homotetramer [21] that specifically binds two sites on the DNA, each site containing 13 base pair recognition sequences [22,23,24]. Our earlier single molecules studies revealed the arrangement of the DNA duplexes in the SfiI synaptosome [14], which corroborates with crystallographic data for the SfiI synaptosome [24]. We used HS-AFM to directly visualize concerted cleavage of DNA in the synaptosome by SfiI [16]. Recently we characterized the efficiency of the formation of looped SfiI synaptosomes depending on the distance between the cognate sites on the DNA [17]. 

In the current report, we applied AFM, including time-lapse HS-AFM, to visualize the dynamics of SfiI-DNA complexes. Experiments were performed in presence of Ca^2+^ cations, which prevents DNA cleavage. The HS-AFM studies showed that SfiI utilizes sliding, jumping, and dissociation-association pathways for the site-search when it binds to DNA through one DNA binding site only. However, SfiI employs a threading mechanism, previously discovered for the EcoRII system [20], in which the protein binds to one specific site and translocates DNA via another binding site, resulting in the change of the loop size. We also identified another site-search pathway, which we termed site-bound segment transfer. In this site-search process, the protein binds to one specific site and jumps, together with the bound DNA, to different segments on the same DNA, resulting in an increase or decrease of the loop size through transient interactions. These data led us to the model for the site-search process describing all known pathways.

## 2. Results

### 2.1. Specific SfiI-DNA Synaptic Complexes with Different Loop Sizes 

We used a 1036 bp DNA substrate containing three recognition sites for SfiI (Figure 1) to characterize the search mechanism of SfiI. Assembly of intramolecular synaptic complexes on this substrate can lead to the formation of SfiI-DNA complexes with specific loop sizes of 254 bp, 532 bp, and 786 bp [17]. AFM images of each complex are shown in Figure 2A. The protein appears as a bright feature connecting two recognition sites on the DNA substrate. According to our previous studies [17], these are highly specific synaptic complexes with high stability compared with non-specific complexes. 

Images in Figure 2A were captured from dry samples. However, similar specific complexes were detected in HS-AFM studies, in which images were acquired in buffer during continuous scanning of the sample. Examples of specific complexes taken from Appendix A are shown in Figure 2B. Plate (i) and Appendix A correspond to the assembly of a complex with a 254 bp loop that remained stable during the observation time. Similar results were obtained for a 532 bp loop (plate ii and Appendix A) and a 786 bp loop (plate iii and Appendix A). The looped complexes formed by SfiI are the key features of synaptosomes, and assembly of a specific looped complex is the final step for the site-search process. 

### 2.2. Dynamics of Synaptosomes: Threading Pathway

We have previously shown, for the EcoRII restriction enzyme [20], that looped complexes can change loop size without protein dissociation. The main feature of such dynamics is that DNA translocates through one protein binding site, whereas another remains bound to a specific site. We termed this dynamic process as the threading pathway. Figure 3 illustrates that SfiI is capable of this search mechanism. The full dataset is provided as Appendix A, and a few frames are shown in Figure 3B. A small loop is initially formed near one end of the DNA substrate (frame 1) and remains unchanged between frames 1 and 8. The loop then grows, as shown in frame 9, while the length of the short DNA flank remains unchanged. We measured the contour length of DNA and separated the measurements into the length of each flank, from the DNA ends to the protein position, and the size of the loop. The results of these measurements are presented in Figure 3C. One flank (red line) remains stable (112 ± 1.7 bp standard error(SE))over the observation window and corresponds to the position of a specific site on the substrate (Figure 1). The other flank decreases in size between frames 8 and 9 from 626 bp to 346 bp. These changes are consistent with an increase in loop size from 273 bp to 557 bp, resulting in the assembly of a specific loop. Thus, these data demonstrate that the protein, while occupying one specific site, is able to translocate via the other DNA binding site without dissociation from the specific site, which is the threading pathway for site-search.

Threading pathway leading to a smaller loop was also observed, and these data are shown in Figure 4. Initial frame with the loop size of ~450 bp is shown in Figure 4A. A few frames from Appendix A are shown in Figure 4B and the results of the contour lengths measurements for the flanks and loop sizes are plotted in Figure 4C. The 450 bp loop remains stable for 23 frames. The loop then shrinks at frame 24, decreasing gradually between frames 24 and 34. The loop size increases at frame 40, but the loop is not stable and fluctuates in size until the protein dissociation at frame 61. The contour length measurements for different DNA segments in Figure 4C show that the length of the short flank 104 ± 0.7 bp (SE), remains constant over the entire observation period. The other flank changes in size, and these changes correlate with the change of the loop size. The loop size decreases from ~450 bp to 350 bp at frame 24 and continues, gradually decreasing to ~200 bp at frame 34. The loop size gradually increases between frames 40 and 50, followed by a decrease of the loop size and protein dissociation at the end of the observation period. 

### 2.3. Site-Bound Segment Transfer

The HS-AFM characterization of looped SfiI-DNA synaptosomes revealed another site-search pathway, shown in Figure 5 and Appendix A. Figure 5A shows the initial frame of the movie and highlights the loop and DNA flanks. Figure 5B shows traces of the DNA to simplify the visual presentation of the images; raw images are shown in Appendix A. The traces demonstrate that a specific synaptic complex, seen in frame 1, releases the distant recognition site between frames 2 and 3, and binds to another distant DNA segment, forming a transient loop between frames 3 and 9. SfiI then binds stably to a specific site, undergoes several transient interactions, and assembles a loop in frame 15. The loop eventually dissociates and the protein dissociates from the DNA (frame 24). We termed this pathway site-bound segmental transfer, as it resembles the intersegmental transfer pathway for the single site-search process but with the protein remaining bound to one site while searching for the other[8]. 

We measured the sizes of the loops and the length of the DNA flanks for the site-bound segmental transfer process, shown in Figure 5C. Arm 1 of the DNA (red arrow) remains stable (108 ± 0.9 bp (SE)) over the observation period and corresponds to the position of the specific site on the substrate (scheme in Figure 1). The initial loop size (green line) corresponds to the 256 bp specific synaptic loop. After the loop dissociation (dashed grey line), a transient ~680 bp loop is formed without protein dissociation from the specific site at 108 bp. Between frames 8 and 12, the loop shrinks slightly, from 677 bp to 645 bp. Finally, another site-bound transfer occurs (frame 15), leading to a loop close to the initial size (266 bp). Taken together, these observations indicate that SfiI uses a site-bound segment transfer mechanism to search the DNA substrate while being bound to one recognition site. 

### 2.4. Sliding and Jumping Pathways

We also characterized SfiI dynamics after the loop dissociation. One such event is shown in Figure 6, with the complete dataset assembled as Appendix A. A few frames are shown in Figure 6B. The initial looped complex in frame 1 dissociates after 12 consecutive frames, following which the protein translocates along the DNA duplex. As a result, by the end of the observation (frame 39), the protein translocates from the point of loop dissociation as a tetramer to a distant site on the same DNA molecule and dissociates from the DNA. 

To characterize the translocation process, we measured the SfiI position, based on the contour lengths of the DNA flanks, and plotted these data in Figure 6C. The short flank (red line), which corresponds to the location of a recognition site, remains stable (104 ± 0.83 bp (SE) until frame 12. Similarly, the loop remains relatively unchanged during this period (green line, 312 ± 2.8 bp (SE), before dissociating at frame 13. Following loop dissociation, the protein remains bound to the DNA duplex. It translocates along the duplex utilizing sliding and jumping pathways, indicated in Figure 6C, with different colors. The graph also shows that the protein slides over the DNA duplex, covering the distance between ~418 bp and 501 bp. After frame 23, SfiI jumps from position 501 bp to 561 bp. SfiI then slides back between frames 28 and 32, reaching the position at 543 bp. Another change in the protein position results from jumps between frames 33 to 39, measuring a distance of almost 300 bp (from 543 bp to 823 bp). Sliding between frames 40 and 41 results in the change of protein position from 823 bp to 842 bp. Finally, SfiI slides back between frames 42 to 45 (from 842 bp to 802 bp) and dissociates from the DNA at frame 46. 

Similar pathways of SfiI site-search dynamics are shown in Appendix A and illustrated with key frames in Appendix A. The SfiI position from the far end of the DNA was measured, and the graph, Appendix A, shows the sliding of the SfiI to the end of the DNA flank. An interesting phenomenon was also observed, in which SfiI, after dissociation from one DNA flank, jumps and binds to another end of the DNA between frames 25 and 36 before dissociating from the DNA.

## 3. Discussion

Time-lapse HS-AFM allowed us to directly visualize the dynamics of synaptic SfiI-DNA complexes and reveal pathways for the site-search process of this type of protein-DNA complex. When the protein binds DNA and searches for one recognition site, it undergoes sliding and jumping pathways, which is in line with the BHW model proposed for the one-site binding proteins [9,10,11,12,13]. As demonstrated in Figure 6, during the jump, the SfiI tetramer dissociates from the DNA, binds to another site on the DNA, and slides before making another jump. Another similar observation was also recorded (Appendix A). The SfiI tetramer slides along the DNA gradually, jumps as a tetramer, and probes the other end of the DNA before complete dissociation from the substrate. SfiI sliding is limited to translocation over dozens of base pairs for these site-search pathways, whereas the protein can cover hundreds of base pairs in the jumping pathways. These two mechanisms can follow each other, allowing the protein to probe the DNA segment between ~400 bp and ~800 bp, improving the efficiency of the site-search process (Figure 6).

The most critical part of our studies was revealing how a synaptic complex, in which the protein binds DNA through two sites, assembles. Our DNA substrate has three cognate sites for SfiI, so the synaptic complexes form looped DNA complexes with the loop sizes corresponding to the distances between the cognate sites. Their assembly is supported by AFM images (Figure 2). Such synaptic complexes correspond to the most thermodynamically stable SfiI-DNA assemblies, so only looped complexes between the cognates for SfiI are visualized with the AFM of dried complexes (Figure 2A and [17]). However, in liquid, we were able to visualize the dynamics of such looped complexes (Figure 3 and Figure 4). The loops change their sizes, suggesting the translocation of the DNA. Importantly, the translocation occurs via one DNA binding site of the protein, whereas the other site remains bound to the specific site. Such DNA threading occurs without visible dissociation of the protein from the DNA, and the overall translocation distance can be as large as ~300 bp (Figure 3 and Figure 4). The loops can grow (Figure 3) or shrink (Figure 4). An increase in the loop size was observed when the initial loop size was rather small, 273 bp (Figure 3). The decrease in the loop size was observed via threading when the initial loop size was 476 bp (Figure 4). This shows that the initial loop size might influence the site-search direction, resulting in growth or shrinkage of the loop to attain the favorable loop size from an energetic point of view. This can be supported with the model proposed in our recent publication [17], where our model predicted that specific loop sizes around 300-350 bp were energetically favorable.

In addition to the threading pathway, one SfiI DNA-binding site can dissociate from the DNA and bind another DNA segment transiently, which can increase or decrease the loop by several hundred base pairs. This dynamic is illustrated in Figure 5. We termed this pathway the site-bound segmental transfer. During this site-search process, transient looped complexes are assembled before the formation of the synaptic complex. Assembly of the transient complex can stabilize the protein-DNA complex, preventing the protein dissociation from the specific site. The transient loop formation thus facilitates the search process either by threading or site-bound segment transfer, resulting in the formation of synaptic protein-DNA complexes.

Altogether we have observed three events for the threading pathway, of which two are presented in Figure 3 and Figure 4. We have observed two events of site-bound segment transfer, of which one event is presented in Figure 5. In addition, we have observed two events of sliding, one of which is presented in Appendix A, and one event of sliding combined with jumping was presented in Figure 6. These findings, together with our previous publication on dynamics of the synaptosomes assembled by the EcoRII restriction enzyme [20], led us to the model for the assembly of synaptic SfiI-DNA complexes shown in Figure 7. The protein binds to the DNA substrate (Figure 7A) and searches for the specific site utilizing sliding, jumping, or site transfer mechanisms (Figure 7B,C). After such assembly, the protein binds DNA at another site, utilizing the site-bound transfer mechanism to make a loop (Figure 7D). Such looped complexes then search for the second specific site utilizing the threading and site-bound segment transfer pathways to assemble the synaptosome looped complex (Figure 7E).

## 4. Materials and Methods

### 4.1. DNA Substrate

The DNA substrate was adopted from our previous study [17]. Briefly, PCR was performed on a pUC19 plasmid (Bio Basic, Markham, ON, Canada) containing an 885 bp DNA segment with three SfiI cognate sites (GGCCTCGAG-GGCC [25]), to obtain the final DNA construct with a length of 1036 bp. The PCR product was run on a 1% agarose gel, and product bands were excised and purified using the Qiagen DNA gel extraction kit (Qiagen Inc., Valencia, CA, USA). Quantification of the purified DNA was performed using absorbance at 260 nm on a NanoDrop spectrophotometer (NanoDrop Technologies, Wilmington, DE, USA). Sanger sequencing was performed to confirm the sequence of the construct and additional verification was performed by restriction digest reaction to confirm all three sites. The final purified DNA substrate consists of three SfiI sites with two flanks of 113 bp (left in Figure 1) and 98 bp (right in Figure 1). The three sites are separated by a distance of 254 bp between the first and second sites and 532 bp between the second and third sites, bringing the distance between first and third sites to 786 bp.

### 4.2. SfiI-DNA Complex Assembly 

Complex assembly was performed as described in our previous study [17]. Briefly, the reaction mixture consisting of 1 μL of 10× buffer A [10 mM HEPES (pH 7.5), 50 mM NaCl, 2 mM CaCl_2_, 0.1 mM EDTA], 2 μL of DNA substrate (86 ng/μL), 1 μL 1 mM DTT, 1 μL SfiI enzyme, and 5 μL di-water was incubated for 15 min at room temperature, diluted through serial dilutions, and deposited on APS-functionalized mica [17] for AFM imaging. SfiI restriction enzyme with low BSA content (20 units/μL) was purchased from the New England Biolabs (Beverly, MA, USA).

### 4.3. Sample Preparation for the HS-AFM Experiment

HS-AFM experiments followed our earlier protocol for protein-DNA complexes [26,27]. Briefly, freshly cleaved mica was functionalized with APS and used immediately for sample deposition. Such a procedure produces a surface where DNA binds weakly, allowing for segmental mobility of the DNA and protein-DNA complexes [26,27]. Appendix A demonstrates the dynamics of the substrate DNA on the APS-mica surface, snapshot from the movie is presented in Appendix A. 

For characterization of the SfiI-DNA complex assembly, 2 μL of the diluted reaction mixture was deposited on the APS-mica. The mica sample was then placed in the HS-AFM cell containing reaction buffer. Imaging was carried by continuously scanning an area on the mica surface, with a typical scan size of 300 nm × 300 nm and scan rate of 600 ms/frame.

To observe the protein dynamics of the SfiI-DNA complexes very gentle, i.e., low tip-sample force, was used during imaging of the sample. This was possible due to low amplitude setpoint as well as use of dynamic PID [28], resulting in the protein being able to move freely in any direction, even against the scan direction or perpendicular to it.

### 4.4. Data Analysis

Analysis was performed using FemtoScan software (Advanced Technologies Center, Moscow, Russia) for the individual frames as described earlier [17,26]. The contour length analysis for the DNA was performed, and the histogram with a single Gaussian function fit is provided in Appendix A. DNA length to base pair conversion factor (0.335) was obtained by dividing the experimentally determined DNA mean length in nanometers by the known DNA length in bp. Loop size of the SfiI-DNA complexes was measured by tracing a line starting from the center of the SfiI tetramer (bright globular features on the AFM images, Figure 2) along the length of the DNA loop back to the center of the tetramer. Translocation events were measured similarly by tracing the DNA filament from the nearest flank end to the center of the SfiI tetramer. Due to the protein size and the temporal resolution, it is plausible that the measurement of the protein position is subject to slight variation.

## 5. Conclusions

The SfiI site-search process was directly visualized using time-lapse HS-AFM. A model for the formation of synaptic protein-DNA complexes is presented based on the data obtained in this study. According to the model, SfiI utilizes threading and site-bound segment transfer pathways to assemble the synaptosome. During the site-search, transient loops are formed, facilitating the search of another specific site. In addition, we have observed the sliding and jumping of SfiI, which the protein utilizes to initiate the site-search process. The contribution of the threading pathway in the site-search process is slightly greater than the site-bound segment transfer pathway in our observation; however, a combination of both site-bound segment transfer and threading facilitates the assembly of the protein-DNA synaptic complex.

## Figures and Tables

**Figure 1 ijms-23-00212-f001:**
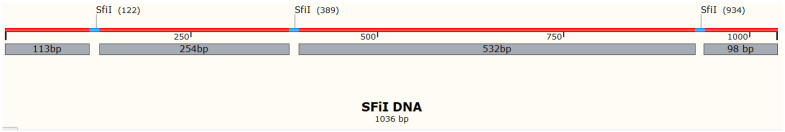
Schematic of the DNA construct with three SfiI recognition sites. The SfiI sites are represented as blue horizontal bars. Numbers below the red bar indicate base pair distance from the left.

**Figure 2 ijms-23-00212-f002:**
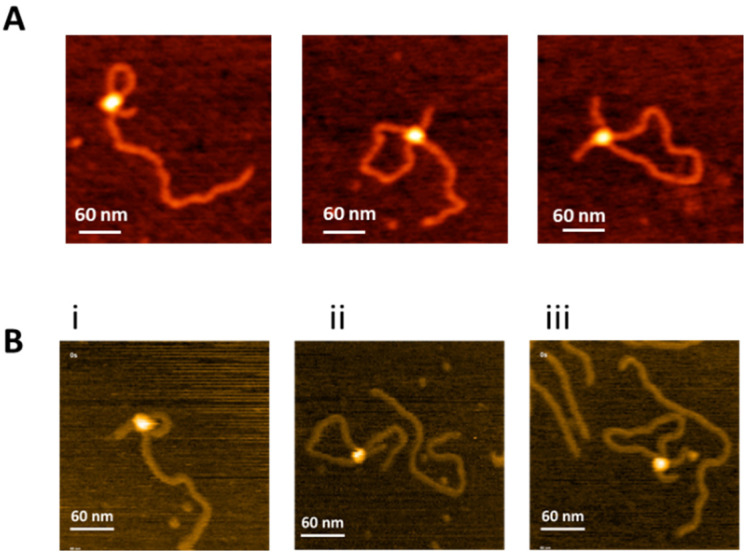
Specific SfiI-DNA synaptic complexes with different loop sizes. (**A**) Left to right, dry AFM images of SfiI-DNA synaptic complexes with loops sizes of 254 bp, 532 bp, and 786 bp, respectively. (**B**) Initial frames of HS-AFM movies (Appendix A) showing the SfiI-DNA synaptic complexes with 254 bp(**i**) 532 bp (**ii**) and 786 bp (**iii**) loops.

**Figure 3 ijms-23-00212-f003:**
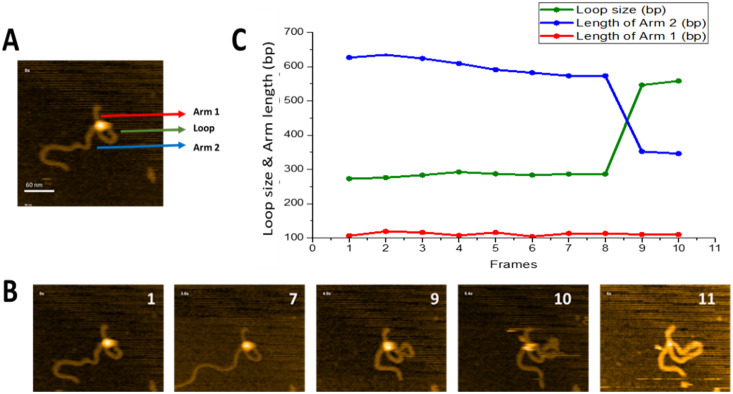
Threading pathway of SfiI site-search leading to loop size increase. (**A**) Initial frame of Appendix A. SfiI tetramer is the bright feature at the cross-over of the DNA loop. The loop is highlighted with a green arrow, the small flank of the DNA (Arm 1) is highlighted with a red arrow, and the long DNA flank (Arm 2) is highlighted with a blue arrow. (**B**) Progression of the loop size changes versus time; numbers indicate frames. The loop remains unchanged between frames 1–7, before growing in frame 9. The new larger loop remains over a single frame before SfiI dissociates in frame 11. (**C**) The graph shows the change in the loop size and DNA arm lengths (in base pairs). The change in loop size and changes in the DNA Arm 1 and Arm 2 lengths are depicted in green, red, and blue, respectively.

**Figure 4 ijms-23-00212-f004:**
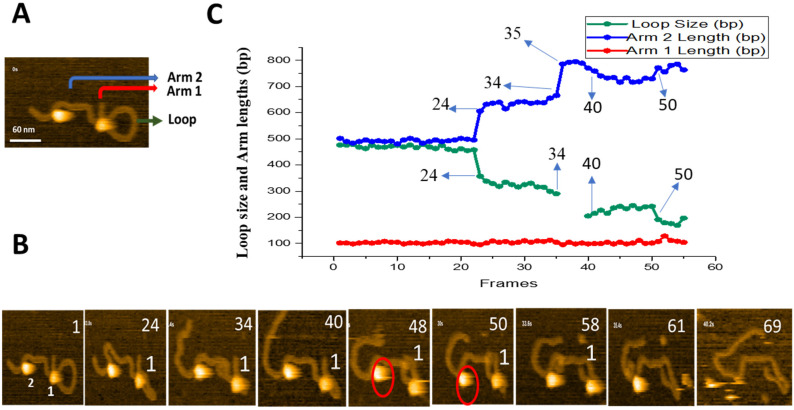
Threading pathway of SfiI site-search leading to loop size decrease. (**A**) Initial frame of Appendix A. SfiI tetramer is seen as a bright feature at the center of the loop structure. The loop structure is indicated with a green arrow, the small arm of the DNA (Arm 1) is labeled with a red arrow, and the long DNA arm (Arm 2) is labeled with a blue arrow. (**B**) Frames with key events. SfiI proteins are labeled as 1 and 2; protein 1 is involved in synaptic assembly. (**C**) Change in DNA contour lengths in bp over time. The green line shows the change in loop size, the red line depicts the change in Arm 1 length, and the blue line shows the changes in Arm 2 length. Other pathways have been observed, and the results are described in sections below.

**Figure 5 ijms-23-00212-f005:**
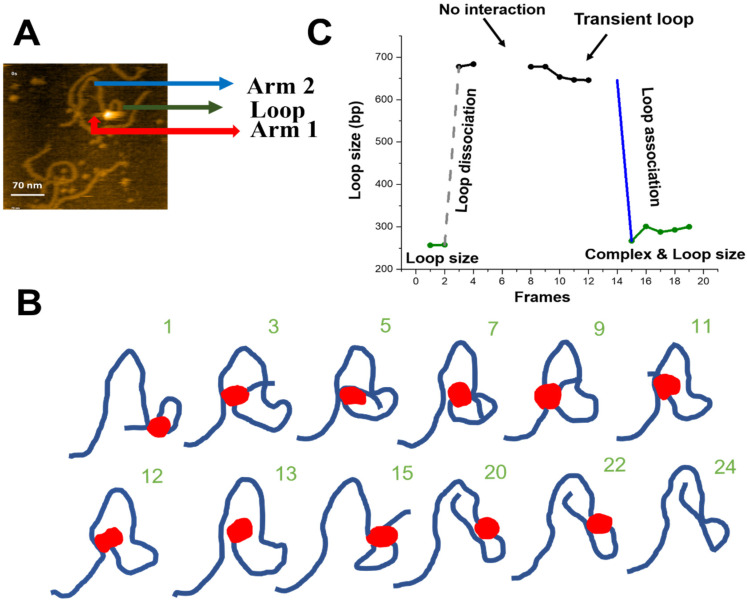
Site-bound segment transfer mechanism of SfiI. (**A**) Initial frame of Appendix A. The sfiI tetramer is the bright feature at the center of the loop structure. The loop structure is indicated with a green arrow, the small DNA arm (Arm 1) is shown by a red arrow, and the long DNA arm (Arm 2) is labeled with a blue arrow. (**B**) Traces of the synaptic complex from selected HS-AFM movie frames; numbers indicate frames. Raw images are shown in Appendix A. (**C**) Loop size change in bp ± SE versus frame number. The green indicates the loop size of the synaptic complex. Grey indicates the dissociation of the loop complex. Black indicates transient loops.

**Figure 6 ijms-23-00212-f006:**
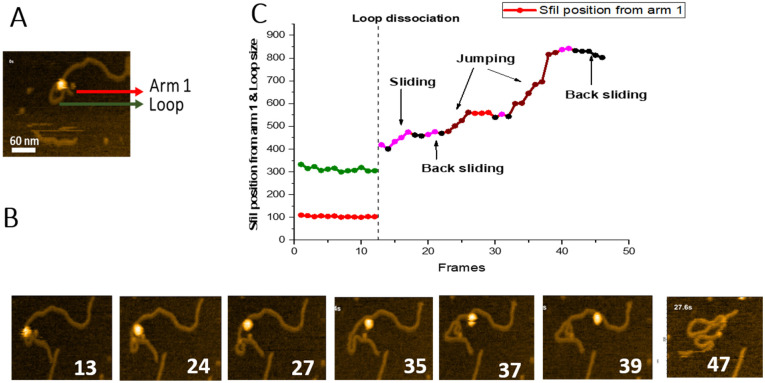
Sliding and jumping of SfiI tetramer after loop dissociation. (**A**) Initial frame of the Appendix A. SfiI tetramer can be seen as a bright feature at the center of the loop structure. The loop is indicated with a green arrow, the small DNA arm (Arm 1) is labeled with a red arrow. (**B**) HS-AFM movie frames with important events between frames 1 and 46; numbers indicate frames. The frames show dissociation of the complex and translocation of the SfiI tetramer along the DNA. (**C**) Graph showing the position of the SfiI tetramer along the length of the DNA measured in base pairs (bp) ± SE. The blue line indicates the length of the DNA Arm 1. The vertical dotted line indicates loop dissociation. The different events observed during were indicated with different colors, as shown in the graph.

**Figure 7 ijms-23-00212-f007:**
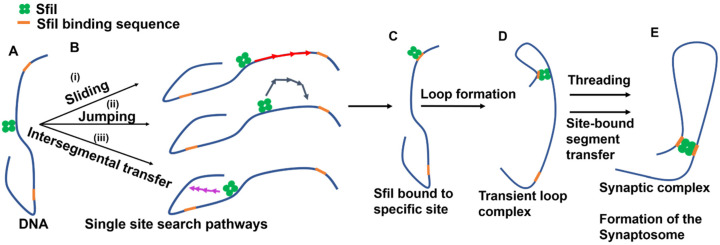
Model for the formation of the synaptosomes by SfiI. (**A**) Initial binding of SfiI (green) to the DNA with two SfiI recognition sites (orange). (**B**) Different site-search pathways. (i) sliding, (ii) jumping, and (iii) intersegmental transfer utilized by the proteins in search fora single specific site. (**C**) SfiI bound to specific site on the DNA. (**D**) Formation of transient loop complexes due to site-bound transfer or DNA diffusion. (**E**) Formation of the synaptic complex after utilizing the threading and or site-bound segment transfer pathways.

## Data Availability

Raw data can be found on the following link: https://www.transfernow.net/en/dltransfer?utm_source=20211219X6u9F0yb&utm_medium=dCuwwKre

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
