# Peer review of "Site-Search Process for Synaptic Protein-DNA Complexes"

_ijms, 2021, doi:10.3390/ijms23010212_

Round 1

Reviewer 1 Report

In this manuscript the authors report their observations of the dynamics of complexes formed between  duplex DNA constructs and  a restriction enzyme SfiI that binds two specific sites on the DNA forming synaptic structures. They use the new, cutting edge, high-speed AFM imaging (HS-AFM) technique to capture live complexes under liquid conditions which allowed them to directly observe different pathways of the search process by SfiI. This is a very impressive work demonstrating again the unique power of this microscopy to resolve single-molecule events which are highly relevant to the enzyme mechanism. I am convinced this work will be appreciated by many researchers in the field of DNA enzymology and AFM studies.

This study must have been very challenging technically, so I do not expect a huge number of independently recorded movies to support the observations and classifications of different pathways, yet I would suggest that the authors comment on this and mention in the manuscript how many movies were captured for each pathway.

They authors provide positions of the tetrameric protein on the DNA with the error on the order of 3-9 base pairs, which is very small considering the resolving power of AFM. The comment on this would be appreciated and the number of observations for supporting a given average position should be provided.

Finally, the positions of the first and third SfiI recognition sites on the DNA construct are approximately equidistant from the respective DNA ends, which makes the absolute identification of these sites difficult. Does it matter for this study?

Author Response

file attached

Reviewer 2 Report

Although it is interesting to visualize the movements of DNA-binding proteins, there are just a few observations without statistics. This undermines any conclusions about routine modes of searching and the authors should make this clear in the presentation of the model.

“Synaptosomes” is a term used to refer to mechanically sheared and resealed nerve terminals. Perhaps “synapses” would be a better choice of terminology although the adjoining segments are not chromosomes.

In Figure 3 the loop size changes between frames as the protein apparently switches binding sites. This movement is abrupt with no intermediate states shown. How can this be unequivocally interpreted as “threading”?

Supplementary files should include the data plotted in the figures.

There is no discussion of the allostery between binding sites in SfiI. Previous research suggests that binding at one site triggers changes at the other site, and that mutants can disrupt this coordination (Bellamy SR, Milsom SE, Kovacheva YS, Sessions RB, Halford SE. A switch in the mechanism of communication between the two DNA-binding sites in the SfiI restriction endonuclease. J Mol Biol. 2007 Nov 9;373(5):1169-83. doi: 10.1016/j.jmb.2007.08.030. Epub 2007 Aug 21. PMID: 17870087; PMCID: PMC2082129.). A discussion of how such allostery relates to the “site-bound” threading and segment transfer search modes must be included.

There is no discussion of how the AFM tip might produce movement by the enzyme. Was movement sensitive to scanning? Did movement persist even during pauses in scanning? Did protein movement correlate at all with the slow scan direction?

In Figure 6 the region labeled “sliding” has almost the same slope as the first region labeled “jumping”. How were these modes of movement distinguished?

In Figure 4 there are two SfiI tetramers but which one is related to the measurements is not indicated. This should be clarified.

In Figure 4 the more stable loop sizes do not correspond to those in the designed DNA. Is this a surface artifact or does it indicate something about the conformation of the “searching” binding site when the other site is bound?

20 of the 35 citations are work by the senior author. This may not be justified. For example, references 29-34 are cited for information about the protocol. If the protocol is contained within one or a subset to these, the number of citations should be reduced. If not, perhaps an updated protocol resulting from the series of publications should be included.

Author Response

file attached
